# A Rail-Temperature-Prediction Model Based on Machine Learning: Warning of Train-Speed Restrictions Using Weather Forecasting

**DOI:** 10.3390/s21134606

**Published:** 2021-07-05

**Authors:** Sunguk Hong, Cheoljeong Park, Seongjin Cho

**Affiliations:** School of Mechanical Engineering, Chungnam National University, 99 Daehak-ro, Yuseong-gu, Daejeon 34134, Korea; hsu12375@gmail.com (S.H.); pffiro@gmail.com (C.P.)

**Keywords:** intelligent transportation system (ITS), machine learning, rail temperature, buckling, XGBoost, structural health monitoring

## Abstract

Predicting the rail temperature of a railway system is important for establishing a rail management plan against railway derailment caused by orbital buckling. The rail temperature, which is directly responsible for track buckling, is closely related to air temperature, which continuously increases due to global warming effects. Moreover, railway systems are increasingly installed with continuous welded rails (CWRs) to reduce train vibration and noise. Unfortunately, CWRs are prone to buckling. This study develops a reliable and highly accurate novel model that can predict rail temperature using a machine learning method. To predict rail temperature over the entire network with high-prediction performance, the weather effect and solar effect features are used. These features originate from the analysis of the thermal environment around the rail. Precisely, the presented model has a higher performance for predicting high rail temperature than other models. As a convenient structural health-monitoring application, the train-speed-limit alarm-map (TSLAM) was also proposed, which visually maps the predicted rail-temperature deviations over the entire network for railway safety officers. Combined with TSLAM, our rail-temperature prediction model is expected to improve track safety and train timeliness.

## 1. Introduction

Rail temperature is important for rail safety. High rail temperature is a direct cause of buckling on railway tracks. Buckling is the result of excessive deformation in a high rail-temperature environment. When severe, it can derail the train. Although train derailment by buckling is an infrequent event, it causes catastrophic casualties of human lives and property [1,2,3,4]. Moreover, the air temperature, which is closely related to rail temperature, is continuously increasing under global warming effects, further enhancing the risk of buckling.

The recent expansion of high-speed trains has increased the demand for continuous welded rails (CWRs), which reduce vibration and noise to offer a comfortable riding experience. Unfortunately, CWRs are also vulnerable to buckling because they are welded together and lack the space to expand. Therefore, controlling the buckling and monitoring the rail temperature is imperative for railway safety management. To prevent buckling-induced train derailment, current railway companies issue train-speed-limit orders based on the rail temperature, which is monitored in real time.

Train-speed-limit orders are directly related to the track safety and train timeliness. If the orders for preventing track buckling (for example, limiting the train speed or spraying the tracks with water) are implemented without planning, the sudden adjustment of the train-operating schedule might cause bottlenecks or traffic congestion. For example, during the hottest July day on record in the UK, there were 12,800 heat-related delay minutes and an additional 23,700 min caused by unplanned train-speed restrictions [5,6]. If the rail temperature in an area can be predicted beforehand, schedulers can preplan the speed limit of the train, adjust the intervals between trains, or cool the rail by water spraying, thereby improving train timeliness and track safety. To achieve this goal, researchers have developed rail-temperature-prediction models (RTPMs) that predict the rail temperature based on the local weather conditions around the rail. 

The previous RTPMs are classified into three types: empirical equation-based models, multivariate regression models, and thermal analysis models [7,8,9,10,11,12,13].

RTPMs based on empirical equations simply predict the rail temperature as a linear function of air temperature. These models are easy to use because they predict the rail temperature from a single parameter, but deliver lower performance (*R*^2^ = 0.9021, root mean square error (RMSE) = 5.866 °C) than other RTPMs [8,9].

Multivariate regression RTPMs predict the rail temperature not only from the air temperature, but also from other weather conditions such as wind speed and cloud cover. Such RTPMs generally outperform those based on empirical equations. Wu et al. developed a multivariate regression RTPM called the Bureau of Meteorology (BoM) prediction equation (1–24 h), which delivers the highest performance to date (*R*^2^ = 0.9630, RMSE = 2.560 °C) but requires 24 features. Therefore, this model is applicable only to specific countries or environments, and is not easily generalized [10].

Finally, thermal analysis RTPMs thermodynamically model the environment around the actual rail. Since thermodynamic models consider the laws of nature, they are more generalizable than RTPMs based on statistical methods, which tend to require specific data. However, to guarantee high performance (*R*^2^ = 0.9334, RMSE = 3.799 °C), such models require precise knowledge of the rail properties, such as the reflectance and emissivity properties [7,10,11,12,13,14].

The previous RTPMs are hampered by two limitations. First, their performance in predicting the rail temperature is too low for practical use. In previous RTPMs based on thermodynamic principles (*R*^2^ = 0.9334, RMSE = 3.799 °C), the predicted rail temperature deviated by up to 15 °C from the measured rail temperature [7]. Since the main goal of RTPMs is preventing buckling on hot days, this difference is unacceptable in practical use. Furthermore, the criteria of train speed limits are shifted to a 4–5 °C change in rail temperature, seriously weakening the reliability of RTPMs [15,16].

Second, the previous RTPMs predicted the rail temperature at a single point. This approach is not meaningful because actual rails are continuously interconnected over several hundred kilometers. To develop RTPMs for practical uses, the range of the rail-temperature prediction must be broadened, for example, from small-town connectivity (local networks) to state- or country-wide connectivity (entire network).

The next generation of RTPMs for practical uses will require high performance RTPM and mapping applications, and will quickly present the rail temperatures from small networks to entire networks.

Recently, machine learning approaches such as artificial neural network (ANN), support vector machine (SVM), random forest (RF), and extreme gradient boosting (XGBoost) have attracted great interest by developers of high-performance regression models. Machine learning allows a computer to learn the relationship between the data (the input) and the results (the output). For instance, a regression model that predicts the room temperature and daily maximum air temperature by machine learning delivers higher prediction performance than other types of methods [17,18].

Herein, we propose a machine learning-based rail temperature prediction model (RTPM) with the highest performance to date (maximum *R*^2^ = 0.9984, RMSE = 0.518 °C) that can predict the rail temperature over an entire network. The method, called Chungnam National University RTPM (CNU RTPM), outperforms the previous RTPMs for predicting high rail temperatures (over 40 °C). The CNU RTPM performance is due to the selected features obtained by analyzing the thermal environment around the rail. With these features, the CNU RTPM can predict the rail temperature over the entire network using weather forecast data alone.

Additionally, a structural health-monitoring application, called the train-speed-limit alarm-map (TSLAM), was developed, which shows the deviation of the predicted rail temperature over the entire network, enabling quick searching of danger regions. This application is easily combined with the CNU RTPM. TSLAM is also available worldwide because it uses the global weather forecast data, not merely a particular region’s data.

## 2. Measurement

### 2.1. Measurement of Rail Temperature and Local Weather Conditions

Direct sampling of the actual rail environment during train operation is impractical for safety reasons. Rail temperatures are usually measured indirectly within a measurement station that simulates the environment of the railway installation [7,10,11,13,19]. As in previous work, we collected the rail temperature and weather data every 10 min at a measurement station from August of 2016 to May of 2017 [7,20]. The constructed measurement station consisted of a 500-mm long KS 50n rail, a data acquisition system (DAQ), a weather station, and K-type thermocouples. The station was installed at a low-traffic site of Chungnam National University (CNU). CNU is located in Daejeon in Korea (latitude: 36.36°, longitude: 127.34°). A schematic of the constructed measurement station is shown in Figure 1 and a photograph of the installed measurement system is shown in Figure A1.

The rail of the measurement system was installed on a ballast-and-concrete sleeper oriented in the south–north direction, which is minimally influenced by shadow. As the actual rail is shadowed by mountains, trees, soundproof walls, and other objects, its temperature is lower than that of the test rail [21]. By minimizing the influence of shadows, a higher temperature is guaranteed in the measurement system than in the shaded area of the real system. Since the purpose of this study was to improve the safety of train operation, we deliberately constructed a conservative measurement system. A model based on conservative measurement data makes conservative predictions.

### 2.2. Measurement of Local Weather Conditions 

Around the world, weather forecasts report the air temperature, relative humidity, rainfall amount, wind speed, and cloud cover. Using these weather factors as features, we can construct a globally available rail-temperature-prediction model. As confirmed in previous studies, the weather factors provided by weather forecasts are decisive predictors of rail temperature. Indeed, a conventional empirical RTPM predicts the rail temperature from the air temperature alone [8,9]. The air temperature and rail temperature are highly correlated.

The weather station (Vantage Pro2, Davis, CA, USA) installed in the measurement system measures the local weather data (air temperature, relative humidity, rainfall amount, and wind speed). The measured local weather data are transmitted to the DAQ. The measured environmental data were acquired once at every 10-min intervals, because it takes about 10-min to achieve thermal equilibrium in the rail measurement system [7]. The rainfall amount is difficult to measure at a specific moment, so it was obtained by summing the values measured over 10 min. The cloud cover, which cannot be measured by the measurement system, was borrowed from the data of the Korea Meteorological Administration (KMA), located 2.4 km from the measurement station.

### 2.3. Measurement of Rail Temperature 

To develop our novel RTPM, measuring the rail temperature in summer when rail buckling usually occurs is needed. However, in summer, the internal and surface rail temperatures can differ by as much as 7 °C, because the sun rises to a higher altitude than in winter, and the rail surface receives a large amount of solar radiation [7,19,22]. Such deviations in rail temperature constitute a noise in the predictive regression model. We overcome this problem by setting a representative point as follows.

Recently, we showed that the deformation of the KS 50N rail at 74 mm from the bottom of the rail represents the average deformation of the whole rail [20]. In this study, the rail temperature at the point of average deformation of the KS 50N rail was taken as the representative temperature of the whole rail. To measure the rail temperature at this point, K-type thermocouple probes were inserted by drilling.

## 3. Feature Study: Feature Selection Based on the Thermal Analysis

### 3.1. Modeling of the Thermal Environment around the Rail

The rail is exposed to complex conduction, convection, and radiation processes that depend on the weather factors (such as the air temperature, wind speed, and cloud) and solar factors (such as solar irradiance) (see Figure 2) [7,11,12]. RTPMs based on the thermal analysis typically predict the rail temperature by balancing the heat transfers [7,11,12,13]. The energy balance equation is given as
(1)$$\dot{E_{\mathrm{sun}}}-\left(\dot{E_{\mathrm{conv}}}+\dot{E_{\mathrm{rad}}}\right)=0,$$
where $\dot{E_{\mathrm{sun}}}$ is the heat flux of the global solar irradiance and $\dot{E_{\mathrm{conv}}}$ and $\dot{E_{\mathrm{rad}}}$ are the heat fluxes of convection and radiation, respectively.

$\dot{E_{\mathrm{sun}}}$ is a major determinant of rising rail temperature and must be modeled in detail [23]. When solar irradiance moves through the atmosphere, it is partially lost to absorption, reflection, and diffusion by moisture, dust, and clouds [24]. If the RTPM is built without considering the detailed solar irradiance, it may predict high rail temperatures inaccurately during the daytime. The poor performance of high rail-temperature prediction is a fatal weakness because high rail temperatures cause rail buckling, especially during the summer daytime.

$\dot{E_{\mathrm{conv}}}$ is closely related to wind speed [7,13]. Generally, the higher the wind speed, the better the heat exchange between the atmosphere and rail surface, considering that the wind speed is high and the air and rail temperatures are similar.

$\dot{E_{\mathrm{rad}}}$ is closely related to the air temperature above the cloud and rail surface emissivity [7]. Based on the Stefan–Boltzmann law, $\dot{E_{\mathrm{rad}}}$ increases to the fourth power of the absolute temperature [25]. In the study of rail temperatures, $\dot{E_{\mathrm{rad}}}$ is of less importance compared to other heat transfer mechanisms. Typically, rail temperatures range from −10 to 60 °C. In this temperature range, $\dot{E_{\mathrm{rad}}}$ is negligible compared to that of other heat transfer mechanisms.

Previous RTPMs based on the regression analysis focused only on the weather factors and did not properly reflect the relationship between solar irradiance and rail temperature, which is critical for rail-temperature prediction. For precise rail-temperature prediction, the rail–sun relationship (solar-effect features) was treated as important while developing the RTPM based on machine learning. Based on the thermal analysis and preliminary studies, the features were classified into weather (air temperature, rainfall amount, wind speed, cloud cover, and relative humidity) and solar effect (azimuth, altitude, and total solar irradiance (TSI)) features.

### 3.2. Weather Features

Most previous regression analysis-based RTPMs predicted rail temperatures considering only weather conditions [8,9,10]. The Hunt models predicted the rail temperature using a single feature (air temperature), whereas the BoM prediction equation used 24 features, including the temperature of the Earth’s surface and pressure [8,10]. 

When building machine learning-based RTPMs, including many weather-features account for various weather situations, but risks the curse of dimensionality and increases the difficulty of data acquisition. To avoid these problems, the relevant features mainly affecting rail temperature must be selected from several weather factors.

Among the weather features, we selected the air temperature, rainfall amount, wind speed, cloud cover, and relative humidity. The effects of the above weather features on the rail temperature were reported in previous studies, and are summarized below.

Air temperature: A change in the air temperature will always change the rail temperature [12]. As mentioned earlier, the simplest RTPM (Hunt models) uses only the air temperature as a feature [8].

Rainfall amount: Moisture is crucial in suppressing the maximum rail temperature because of the high specific heat of water (4.184 J∙g^−1^∙K^−1^)) [13].

Wind speed: As mentioned earlier, wind speed is closely related to convection. When the wind speed increases, the difference between the air temperature and rail temperature decreases. This effect is enhanced in summer, when the difference between the rail and air temperatures is already large [11,13].

Cloud cover: Cloud cover is closely related to solar irradiance. On cloudy days with high cloud cover, the portion of lost solar irradiance increases and the ratio of diffuse horizontal irradiance (DHI) to global solar irradiance (GHI) increases accordingly [13].

Relative humidity: A change in the relative humidity slightly changes the rail temperature [13]. In Chapman’s model, the relative humidity is assumed to be closely related to the minimum rail temperature [13].

The suggested weather features were measured as described in Section 2 and are easily obtainable worldwide because they are common components of global weather forecasts. By virtue of the weather features, the CNU RTPMs built by the suggested features can predict the rail temperatures over an entire network. 

### 3.3. Solar Effect Features

The sun is the main heat source of the rail and causes a high rail temperature during the daytime. Therefore, the influence of the sun must be considered in the rail temperature prediction. To predict the rail temperature over the entire network, we here define the solar effects (azimuth, altitude, and TSI) as new features that can be simply calculated in terms of time (year, month, day, hour, minute, and second) and location (latitude and longitude). By using the solar effect features in the RTPM-building, the CNU RTPM can account for the influence of solar irradiance on rail temperature. The solar effect features release the temporal and location dependencies of the RTPMs, because they are physical variables that change in time and space. Since the rail temperature has increased under global warming effects, RTPMs that depend on a specific time and location cannot predict the rail temperature in the coming decades or at other locations. In future RTPMs that predict the rail temperatures of entire networks, the solar effect features are essential.

#### 3.3.1. Modeling of Solar Irradiance

The GHI is closely related to rail temperature, so it must be considered in the rail-temperature prediction. The measured rail temperature and GHI data during the summer of 2016 at the measurement station are compared in Appendix B. Note that the rail temperature and GHI exhibited similar temporal dynamics (Figure A2a) and were positively correlated (Figure A2b; *R*^2^ = 0.6870). The phase difference in Figure A2a is attributable to the heat transfer delay caused by the heat capacity of the rail. 

Clearly, the GHI is an important predictor of rail temperature and would improve the performance of RTPM. As this variable is not provided in the weather forecast, it was replaced by the TSI on the Earth’s upper atmosphere. The TSI includes the solar power over all incident wavelengths per unit area and changes slowly as the Earth elliptically orbits the sun [25,26]. As the TSI depends on the distance between the Earth and the sun, it can be predicted accurately. The GHI can then be predicted from the TSI and the weather factors [26,27,28,29]. 

The TSI, DNI, and DHI are geometrically defined in Figure 3a. Upon entering the atmosphere, the TSI is scattered and absorbed by weather factors such as humidity and clouds. At noon on a clear day, approximately 25% of the TSI is scattered and absorbed [30].

The DNI is the solar irradiance directly arriving from the sun. The DNI is measured as the flux of the beam radiation through a plane perpendicular to the sun’s direction [23]. 

The DHI represents the solar irradiance scattered by the atmosphere. The DHI is measured on a horizontal surface, assuming that the radiation (excluding the circumsolar radiation) enters from all points in the atmosphere.

The GHI is the total irradiance from the sun on a horizontal surface on Earth [27]. It is computed by summing the DNI and DHI.

In summary, since the TSI is the reduced GHI by weather factors, simultaneously adopting the weather factors and TSI as features in the RTPM produces the same effect as directly adopting the GHI as the feature. The TSI is calculated as described in Appendix C.

#### 3.3.2. Direction of Solar Position

As the Earth is spherical, the TSI enters the atmosphere at an oblique angle, so the GHI (the summed DNI and DHI) varies in each region. The DNI is oriented along the direction of the sunlight, which is defined by its azimuth and altitude. The changing direction of sunlight alters the sites at which the rails receive energy directly from the sun. Therefore, even when the GHI remains constant, the rail temperature changes with azimuth and altitude [7].

In the horizontal coordinate system, the direction of the sun is expressed by its azimuth and altitude, as shown in Figure 3b [30]. The azimuth (*Φ*_sun_) is the angle between the projected vector and the north (N) vector. The altitude (*α*_sun_) is the angle between the sunlight vector and its projected vector on the perpendicular plane. 

The azimuth and altitude are calculated in terms of time (year, month, day, hour, minute, and second) and location (latitude and longitude). The temporal terms are astrophysically meaningful for predicting the rail temperature. If the time is directly applied to the model without any post-processing, the model’s performance is not guaranteed after decades or longer, and its reliability is severely degraded.

In this study, the altitude and azimuth were computed from the local latitude, longitude, and time by the Michalsky’s method [31].

## 4. Building the CNU RTPMs 

CNU RTPMs are based on machine learning and various statistical methods: extreme gradient boosting (XGBoost), support vector machine (SVM), random forest (RF), artificial neural network (ANN), and polynomial regression of order 2 (PR2). Their hyperparameters were optimized by tuning. The performances of the various CNU RTPMs were compared and the best-performing CNU RTPM was clarified. We built the models using the Python libraries: XGBoost, Scikit-learn, random forest, and Tensorflow.

### 4.1. Machine Learning and Statistical Methods 

#### 4.1.1. Extreme Gradient Boosting (XGBoost)

XGBoost is popularly chosen by contestants in data and machine learning competitions, owing to its higher performance than other methods [32,33,34,35,36,37]. XGBoost is suitable for regression and classification problems. At each step, the algorithm generates a weak learner and accumulates it into the total model. If the weak learner follows the gradient direction of the loss function, the learning method is called a gradient boosting machine [37].

XGBoost is an ensemble of classification and regression trees (CART). An ensemble method is a machine learning technique that combines the results of several submodels to determine the best result. Consider a dataset ${D=\{(X}_{i},\;y_{i}):\;i=1\cdots n,\;X_{i}\in R^{m},\;y_{i}\in R$} with n data, m feature spaces, a target value $y_{i}$, and a predicted value ${\hat{y}}_{i}$ Let ${\hat{y}}_{i}$ be the result of an ensemble represented as follows:(2)$${\hat{y}}_{i}=\overset{K}{\underset{k=1}{\sum }}f_{k}\left(X_{i}\right),\;f_{k}\in \;F,$$
where *K* is the number of trees and $f_{k}$ is a function in the functional space F  of all CARTs.

Let *L* be the objective function, which consists of a training loss term and a regularization term as follows:(3)$$L(\phi )=\sum_{i}l\left(\hat{y_{i}},y_{i}\right)+\sum_{k}\Omega \left(f_{k}\right),$$
(4)$$\Omega \left(f_{k}\right)=\gamma T+\frac{1}{2}\lambda {\left\|w\right\|}^{2}.$$

The training loss term l measures the difference between ${\hat{y}}_{i}$ and $y_{i}$, and the regularization term Ω penalizes the complexity of the model to avoid overfitting.

The objective function is optimized by tree boosting of Equation (3). Define ${\hat{y}}_{i}^{t}$ as the prediction of the *i*-th instance at the *t*-th iteration. To minimize the objective function, a term $f_{t}$ is added to Equation (3), giving Equation (5). Equation (5) is then simplified by the Taylor expansion to give Equation (6):(5)$$L^{t}=\sum_{i=1}^{n}l(y_{i},{\hat{y}}_{i}^{t-1}+f_{t}(X_{i}))+\Omega (f_{t})$$
(6)$${\overset{ˇ}{L}}^{t}=\sum_{i=1}^{n}{[g}_{i}f_{t}\left({Χ}_{i}\right)+\frac{1}{2}h_{i}f_{t}^{2}\left({Χ}_{i}\right){]+\Omega (f}_{t}),\;\mathrm{where}g_{i}=\partial_{{\hat{y}}_{i}^{t-1}}l(y_{i},{\hat{y}}_{i}^{t-1}),h_{i}=\partial_{{\hat{y}}_{i}^{t-1}}^{2}l(y_{i},{\hat{y}}_{i}^{t-1}).$$

The loss reduction after the best split from a given node is given by Equation (7). Note that this function depends only on the loss function and the regularization parameter *γ*. Clearly, this algorithm optimizes any loss function that provides the first and second-order gradients [32].
(7)$$L_{split}=\frac{1}{2}\left[\frac{{\left(\sum_{i\in I_{L}}g_{i}\right)}^{2}}{\sum_{i\in I_{L}}h_{i}+\lambda }+\frac{{\left(\sum_{i\in I_{R}}g_{i}\right)}^{2}}{\sum_{i\in I_{R}}h_{i}+\lambda }-\frac{{\left(\sum_{i\in I\;}g_{i}\right)}^{2}}{\sum_{i\in I}h_{i}+\lambda }\right]-\gamma .$$

#### 4.1.2. Support Vector Machine (SVM)

The objective of the SVM is to find the optimal hyperplane in an *N*-dimensional space (where *N* is the number of features) that maximizes the distance between the hyperplane and the nearest data point on each side. The optimal hyperplane should distinctly classify the data points [38]. SVMs are employed in regression and classification problems, and are applicable to both linear and nonlinear data [39].

As in previous studies, our study employs an SVM for predicting the solar generation from weather forecast data. The kernel function was a radial basis function with two hyperparameters: the kernel coefficient γ and the penalty coefficient *C*.

#### 4.1.3. Random Forest (RF)

An RF is a substantial modification of bagging that builds a large collection of de-correlated trees, and then averages them [40]. Bagging is one of the ensemble methods. Like SVMs, RFs are applicable to both linear and nonlinear data in regression and classification problems.

In previous studies, RFs have predicted the daily maximum air temperature from the solar radiation, albedo, latitude, longitude, and other solar-related parameters. We similarly adopt the solar effect features in temperature prediction. Here we optimize the number of variables to be split at each node (*M*_try_) and the number of trees in each run (*N*_tree_) as the hyperparameters.

#### 4.1.4. Polynomial Regression of Order 2 (PR2)

PR2 is a traditional statistical method applied in regression modeling. It is suitable when the independent variable is a quadratic function of the dependent variable [38]. Additional hyperparameter settings are not required. The PR2 can be expressed as
(8)$$y_{i}=\beta_{0}+\beta_{1}x_{i}+\beta_{2}x_{i}^{2}+\cdots +\epsilon_{i},$$
where $y_{i}$ is the dependent variable,$\;x_{i}$ is the independent variable, $\beta_{i}$ denotes the unknown parameters, $\epsilon_{i}$ is an error term, and *i* denotes a row of data.

#### 4.1.5. ANN (Artificial Neural Network)

ANNs are powerful connectionist systems vaguely inspired by biological neural networks [41]. An ANN consists of multilayer perceptrons, and is suitable for solving various problems such as computer vision, speech recognition, and regression analyses [42].

Previously, ANNs were applied to body-temperature prediction in wearable-device studies [43]. In our study, an ANN predicts not the body (interior) temperature but the skin (surface) temperature. Specifically, we employ a feed-forward neural network with a rectified linear unit as the activation function and AdamOptimizer as the optimizer. The learning rate and batch size were 0.05 and 100, respectively. We optimize the number of nodes (one hidden layer) as the hyperparameter.

### 4.2. Hyperparameter Tuning by K-Fold Cross Validation 

The performance of any model based on machine learning largely depends on the value(s) of the hyperparameter(s), which must be determined before running the model. In machine learning, the process of determining the hyperparameter(s) is called “tuning.” [44,45,46]. The suitability of the value range of the hyperparameter(s) tends to depend on the user’s intuition and experience.

In this study, the hyperparameters of XGBoost, SVM, and RF were optimized by K-fold cross validation, which divides the samples into training and test samples. The model was constructed using the training data and verified using the test data. K-fold cross validation simply and effectively checks whether the model has overfitted the training data.

K-fold cross validation divides the samples into k uniform groups and performs k iterations of cross validation. After cross validation based on the constructed hyperparameter values, the optimal combinations of hyperparameters are found using the GridSearchCV function in the Python library Scikit-learn. By default, GridSearchCV uses 3-fold cross validation (*k* = 3); however, applying 5-fold cross validation is conducted to more accurately optimize the hyperparameter combination.

The results of the hyperparameter tuning are provided in Appendix D.

### 4.3. Comparison Methods of Model Performance 

In this study, 70% of the measured data were randomly selected as the training data and the remaining 30% were reserved as the test data. To account for the randomness in distributing the training and test data, the evaluation was performed five times on the test data and individual performances were averaged to obtain the final performance value. As the performances of the training and test data were not significantly different at the time of the experiment, the results of the test data are presented. The performances were described by the mean absolute error (MAE), coefficient of determination (*R*^2^), and RMSE.

MAE is the absolute value of the average error (average difference between the original and predicted values). A lower MAE corresponds to a higher performance of the model. The R^2^ measures how well the predicted values match the original values. Its value ranges from 0 (no correlation between the actual and predicted values) to 1 (perfect correlation between the two values). RMSE is another measure of the average error in the prediction. As MAE is generally used in machine learning and RMSE is conventionally used in RTPMs, both measures are computed in the present analysis. Similar to MAE, a low RMSE implies a high model performance. Conclusively, a good prediction model is characterized by low MAE, high *R*^2^, and low RMSE. MAE, *R*^2^, and RMSE measures are respectively expressed as follows:(9)$$MAE=\frac{\sum_{i=1}^{n}\left|y_{I}\;-\bar{y_{i}}\right|}{n},$$
(10)$$R^{2}=1-\frac{\sum_{i=1}^{n}{\left(y_{i}-\bar{y_{i}}\right)}^{2}}{\sum_{i=1}^{n}{\left(y_{i}-\hat{y_{i}}\right)}^{2}}$$
(11)$$RMSE=\sqrt{\frac{1}{n}\sum_{i=1}^{n}{\left(y_{i}\;-\;\bar{{\;y}_{i}}\right)}^{2}},$$
where *n* is the total number of test data, *y_i_* and *ȳ_i_* are the measured data and the model predicted values, respectively, and *ŷ* is the average output value of the test data.

The features and measurement conditions determine the reliability and versatility of the prediction model. As the model is built using the measured data, a small quantity of measured data will degrade the model reliability. Meanwhile, if the model includes an excessive number of features, it cannot easily adapt to different environments and the measurement system requires many instruments. Accordingly, models with too many features have low versatility.

## 5. Train-Speed-Limit Alarm-Map (TSLAM)

The actual rails are interconnected continuously for several hundred kilometers. Therefore, predicting the rail temperature in a local region is meaningless. Our TSLAM framework, developed by Python, visually presents the predicted rail temperature over the entire network. TSLAM visualizes the predicted rail temperature deviations computed by the CNU RTPMs with the selected features. Using TSLAM, train safety officers can quickly map the rail-temperature deviations and order the train company to limit the train-speed in advance.

The TSLAM can also forecast the rail temperature for up to 64 h based on the weather forecast data alone. This forecasting reduces train delays owing to the imposed train-speed limit because the train company can preadjust the train operation interval. Finally, the TSLAM improves track safety and train timeliness.

The TSLAM operates via a four-step algorithm: data acquisition, data preprocessing, rail-temperature-prediction, and data visualization (see Figure 4). The four steps of the TSLAM algorithm are detailed below.

Step (1): Web crawling of weather forecast data: The KMA divides the Korean peninsula into (5 × 5)-km^2^ grid squares and forecasts the weather in each area for up to 64 h at 3-h intervals. Step 1 of the TSLAM algorithm obtains the local forecast data by web crawling (defined as data extraction from a web page [47]). The obtained data comprise the location and meteorological forecasts in each area. In this step, if the data source of the application programming interface (API) provided by the open weather map (OWM) is changed, the country and its local weather forecast data are newly obtained. Originally, we considered using the Korean local weather forecast of the OWM rather than the local weather forecast of the KMA. However, web crawling the weather forecast data of KMA, which provides a detailed local weather forecast over more regions in Korea than in the OWM, was selected.

Step (2): Data preprocessing: The KMA provides local coordinate data in its own coordinate system, which must then be converted to general-purpose latitudes and longitudes. This step is redundant if the data source is OWM’s API because the OWM directly provides local latitude and longitude data. This step then computes the azimuth and altitude from the latitude and longitude at the forecast time and converts the data unit into units compatible with CNU RTPM.

Step (3): Prediction of the rail temperature using CNU RTPM: This step predicts the rail temperature from CNU RTPMs. CNU RTPM with the highest performance is then installed as the main model. Through this process, data are converted into three-dimensional data of latitudes, longitudes, and predicted rail temperatures.

Step (4): Visualization of the predicted rail temperature: This step displays the transformed three-dimensional data on a map for the user. The displayed image can be expressed in either of the following two modes, Mode 1 or Mode 2, by adjusting the range of the legend. Mode 1 is a contour map showing the predicted rail temperature in all regions of the selected country. Mode 1 compares the predicted rail-temperature deviations in each region but does not clarify the regions in which rails are dangerously deformed. Conversely, Mode 2 shows the regions of high predicted rail temperature at which the train-speed should be limited to ensure safe railway operation. These data quickly inform the train safety officer of the risky sites.

## 6. Results and Discussion

### 6.1. Data Configuration 

The most important goal for building a model based on machine learning is the acquisition of numerous high-quality data. Herein, high-quality data were acquired using the measurement station. The rail temperature was determined at the point where rail deformation represented the average rail deformation. Moreover, features providing a globally usable model were selected.

Measurements were continuously collected over a 10-month period from August 2016 to May 2017, obtaining 35,252 samples. The measured data were the air temperature, relative humidity, wind speed, rainfall amount, and rail temperature. The cloud cover data, which were not measured, were instead borrowed from the data of nearby KMA. The maximum, minimum, average, and standard deviation of the dataset are included in the Appendix A. Korea’s climate is characterized by a large annual range of air temperatures (−10.1–38.0 °C in the collected dataset) and by various weather events, such as monsoon rains and snowfall, which are dynamically generated in different seasons. The Korean weather data are thus considered suitable for creating globally available models. 

The azimuth, altitude, and TSI were calculated by the method presented in Section 3 and added to the final samples. The resulting model features were weather features (air temperature, rainfall amount, wind speed, cloud cover, and relative humidity) and solar effect features (azimuth, altitude, and TSI).

Herein, 70% of the data (24,676 samples) were randomly selected as the training data and the remaining 30% (10,576 samples) were reserved as the test data. Figure 5 is a detailed flowchart of the methodology. 

### 6.2. Feature Importance 

Tree-based machine learning methods such as XGBoost can calculate the importance of features. The feature importance quantitatively evaluates the effect of a particular feature on the model performance. By ranking and comparing the importance of each feature, we can remove the unnecessary features with low feature importance from the model. Such low-importance features can degrade the performance of the model. 

The feature importance values computed by XGBoost are graded in Figure 6. The most important, second-most important, and third-most important features were the TSI, azimuth, and altitude, respectively. Note that these features are the solar effect features reflecting the GHI, which are obtained by simple calculations but largely affect the performance of the model. That is, the solar effect features play an important role in predicting the rail temperature.

### 6.3. Model Performance Comparison

#### 6.3.1. Performance at the Whole Range of Rail Temperatures

Table 1 compares the performances of the previous RTPMs (white) and the newly developed CNU RTPMs (yellow and orange). To compare the models’ performances, the datasets used in the RTPMs should be similar. However, direct performance comparison was difficult because of the restricted data set used in developing previous RTPMs. Thus, the Hunt model, which uses the air temperature as the sole feature, was used to confirm the similarity of the dataset used in this study with that used in other RTPMs. Besides, the Hunt model is common because similar studies have also compared model performance using it [7,10,13]. In a study by Wu (development of the BoM prediction equation), the RMSE of the Hunt 1 model was determined as 6.952 °C, while in this study, it was determined as 5.866 °C. These results disprove the similarity between Wu’s study and the measurement conditions adopted in this research, such as the rail installation environment and climate. Figure 7 shows the predicted and measured rail temperatures from August 9 to August 12 in 2016, when the rails were the hottest. The Hunt 1 model predicts the temperature of the rails to be somewhat higher, while the CNU RTPM–XGBoost predicts it with high accuracy regardless of day or night conditions. 

Among the previous RTPMs, the BoM prediction equation showed the highest performance (MAE = 0.136 °C, *R*^2^ = 0.9630, RMSE = 2.560 °C) over the whole range of rail temperatures. However, the BoM equation requires 24 features, which severely reduces its versatility. Moreover, the BoM equation is constructed from data collected over one month during the Australian winter, casting doubt on its global and seasonal applicability. In contrast, the CNU RTPMs use consecutive data collected over 10 months (including summer and winter), ensuring their reliability over the four seasons. Moreover, the weather factors and solar effects are easily obtainable, ensuring the versatility of the CNU RTPMs. The weather factors are globally available through weather forecasts, and the solar effects are readily calculated at any given time. These selected features enable easy combination of CNU RTPM and TSLAM. 

The highest performer among the RTPMs over the whole range of rail temperatures was CNU RTPM–XGBoost (MAE = 0.008 °C, *R*^2^ = 0.9984, RMSE = 0.518 °C). The RMSE was approximately 2 °C lower in CNU RTPM–XGBoost than in the BoM prediction model, the previously highest performance RTPM.

Accordingly, CNU RTPM–XGBoost was selected as the main model in this study and its hyperparameters were optimized by K-fold cross validation. Adding the solar effect features improved the performance of CNU RTPM–XGBoost over that of CNU RTPM–XGBoost without solar effects, built using the weather data only. 

#### 6.3.2. Performance at the High Rail-Temperature Range (over 40 °C)

The most important practical requirement of RTPM is predicting the high rail temperatures that cause buckling. When predicting rail temperatures over 40 °C (Table 1), the CNU RTPM–RF and CNU RTPM–XGBoost delivered the highest performance among the tested models (CNU RTPM–RF: MAE = 0.191 °C, *R*^2^ = 0.9199, RMSE = 0.927 °C; CNU RTPM–XGBoost: MAE = 0.119 °C, *R*^2^ = 0.9415, RMSE = 0.771 °C). The ability to predict high rail temperatures was conferred by a selected algorithm and proposed features.

Since they adopt the ensemble method, tree-based machine learning algorithms (CNU RTPM–RF and CNU RTPM–XGBoost) are more suitable for high temperature prediction than other machine learning algorithms. Tree-based RTPMs are composed of various submodels constructed under specific conditions. As one of these submodels predicts high rail temperatures, tree-based RTPMs can outperform models with other architectures.

Additionally, we showed that the newly proposed solar effect features further improve the performance of predicting the high rail temperature in RTPM by comparing the performance of the CNU RTPM–XGBoost with or without the solar effect (*R*^2^ = 0.9415 and 0.7243, respectively). 

#### 6.3.3. Raw Error Data of CNU RTPM

The practical applicability of CNU RTPMs was investigated on raw error data. The performances of the CNU RTPMs with the same features are compared in Figure 8. This figure shows the box plots of the errors in the CNU RTPMs, allowing a comparison of the raw error data in each CNU RTPMs. 

The box plots differ in shape; the fence length (distance between the upper and lower whiskers) is shorter in the CNU RTRMs based on machine learning (SVM, ANN, RF, and XGBoost) than in the regression model (PR2). The tree-based RTPMs (RF and XGBoost) yielded the shortest fence lengths around a median of 0 °C.

The error ranged by almost 9 °C in CNU RTPM–PR2, SVM, and ANN, nearly 6 °C in CNU RTPM–RF, and nearly 3 °C in CNU RTPM–XGBoost. A narrow error range guarantees a reliable rail-temperature prediction that minimizes the risk of incorrect prediction.

As confirmed by its short fence lengths and narrow error range (3 °C), the CNU RTPM–XGBoost is the most suitable algorithm in practical situations, because the criteria of train-speed limits are shifted to a 4–5 °C change in rail temperature. This varies from country to country. In Korea, high-speed trains run at 230 km∙h^−1^ when the rail temperature is 55–60 °C, 70 km∙h^−1^ when the rail temperature is 60–64 °C, and are cancelled when the rail temperature exceeds 64 °C [15]. In the UK, the train speed is determined by the stress-free temperature (STF, usually equal to 27 °C in the UK). On well-maintained tracks, the speed limit is 60 mph and 20 mph at rail temperatures of STF + 37 °C and STF + 42 °C, respectively. However, under poor track conditions (inadequate ballast), the temperatures of the 60 mph and 20 mph speed limits reduce to STF + 13 °C and STF + 15 °C, respectively [21]. 

### 6.4. TSLAM with CNU RTPM–XGboost

TSLAM is a structural health-monitoring application for analyzing railway safety using CNU RTPM–XGBoost, the highest-performing CNU RTPM. Using TSLAM, railway safety managers can know the rail temperature in advance or in real-time. Based on the predicted rail temperature, railway safety managers could decide on safety measures, such as limiting the train speed and spraying water.

Figure 9a shows the graphical user interface (GUI) of TSLAM. The user selects the time, country, and map mode through input-selection widgets and obtains the rail temperature predicted by CNU RTPM–XGBoost. The predicted rail temperature is displayed either in Mode 1 (the standard mode showing the predicted rail temperatures in all regions) or Mode 2 (the detection mode displaying the danger zones on the map). The text-display panel reports the risk points demanding rail-temperature management and the train-speed limits recommended by the Korea Railroad Operation Safety Managers Association [15] (see Appendix E). Equipped with this information, railway safety officers can either plan the lowering of the rail temperature by water spraying or adjust the train operation interval in advance.

The main advantage of TSLAM is its worldwide applicability, enabled by including the globally available features in CNU RTPM–XGBoost. A user can select a country through the GUI, and visually observe the predicted rail temperature of that country using the weather forecast provided by the OWM. In other words, users can obtain the rail temperature or warnings from devices installed with TSLAM, independent of location. As an example, Panels (b) and (c) of Figure 9 show the predicted rail-temperature maps of France and the USA, respectively, in Mode 1. In this mode, users can compare the regional deviations in the predicted rail temperatures. In Mode 2 (Figure 9a), the user must know the train driving rules of the selected country.

TSLAM can improve the efficiency of a train system controlled by a safety management system. As mentioned above, TSLAM requires the weather forecast data from the OWM, which are easily accessible. Therefore, operators of a train safety management system can acquire the information required for rerouting or applying derailment-prevention measures at any locality. With this information, they can appropriately adjust the train interval, avoid bottlenecks, and efficiently operate the rail system.

Additionally, TSLAM can support the indirect measurement of the rail temperature over the entire network. Generally, the rail temperature is directly measured using thermocouples; however, installing thermocouples over the entire network results in high installation and maintenance costs. Additionally, because of the safety issue, it is difficult to attach the thermocouples in some networks. However, TSLAM prediction showed nearly the same result as directly measuring the rail. Thus, TSLAM is expected to serve as a supporting system that provides the predicted rail temperature using real-time weather measurement data or weather forecast data.

### 6.5. Limitations and Directions for Improvement

The features of CNU RTPM were designed by using the thermal analysis. The aim was to improve the performance of rail-temperature prediction and to extend its network applicability. However, some relevant features were excluded because of data unavailability and restriction. The particulate matter (PM) is one of the excluded features, which can be incorporated into an improved RTPM in future work.

Recently, PM2.5 (with diameters of 2.5 μm or less) has been shown to decrease the DNI proportion and increase the DHI proportion in the GHI [48]. By definition, the DNI is related to the direction of sunlight and induces a temperature distribution on the cross- section of the rail. In contrast, the DHI is less directional and tends to smooth the temperature distribution on the rail cross-section. In regions with high PM2.5 concentration, this interplay causes a slight difference between the rail temperature predicted by CNU RTPM and the actual rail temperature.

Moreover, sleeper material might be incorporated into an improved RTPM in future work. During the measurements, the rail was installed on concrete sleepers with a thermal conductivity of 0.13 W∙m^−1^∙k^−1^ [49]. Although the installation of concrete sleepers is increasing with the development of high-speed trains, wooden sleepers remain common in many parts of the world. The thermal conductivity of wood is 2.0 W∙m^−1^∙k^−1^ [49,50]. The different thermal conductivities of these two materials will degrade the universal applicability of CNU RTPM. Concrete sleepers with low thermal conductivity will conduct a lower heat flux than wooden sleepers. For this reason, when the solar irradiance is high, a rail installed on wood sleepers will reach a lower temperature than a rail installed on concrete sleepers.

Furthermore, the health status of rails can be integrated into an improved RTPM. The health status of the rail is directly related to its thermal properties, such as solar absorptivity, and mechanical properties, which include hardness and density [7,51]. For example, according to a previous study, solar absorptivity is important for predicting rail temperatures [7]. The solar absorptivity expresses how much solar irradiation affects the temperature change of the rail. The rail surface condition affects the solar absorptivity. Usually, the solar absorptivity of unused rail is 98.7%, whereas that of used rail is 81.1%. This difference comes from the rail surface and paint erosion. Thus, the performance of RTPM can be improved in future works by considering the health status of rail.

## 7. Conclusions

Herein, we developed novel rail-temperature prediction models (CNU RTPMs) and a TSLAM using weather forecast data alone, which can predict the rail temperature over the entire network.

The CNU RTPMs were developed with different machine learning methods using the long-term (over 10 months) measured data from all seasons. Such long-term data collection ensures a reliable model.To improve the prediction performance, the CNU RTPMs combine standard weather features with newly suggested solar effect features. These features originate from the analysis of the thermal environment around the rail. Additionally, they are easily obtained from global weather forecasts and additional calculations on the weather forecast data. Precisely, the solar effect features significantly improved prediction performance at the high-rail-temperature range.In a performance comparison, the CNU RTPM–XGBoost emerged as the best predictor of the rail temperature among the machine learning methods. The proposed CNU RTPM–XGBoost, which delivered higher performance, reliability, and versatility than previous RTPMs, was suggested as a new model for predicting rail temperature over the entire network. The CNU RTPM–XGBoost is applicable worldwide because its features are globally available in weather forecast data. The visualization application, TSLAM, maps the predicted rail temperatures, which assist railway safety officers (if necessary) in planning safety measures. 

We expect that CNU RTPM–XGBoost and TSLAM will significantly improve both train safety and train timeliness.

## Figures and Tables

**Figure 1 sensors-21-04606-f001:**
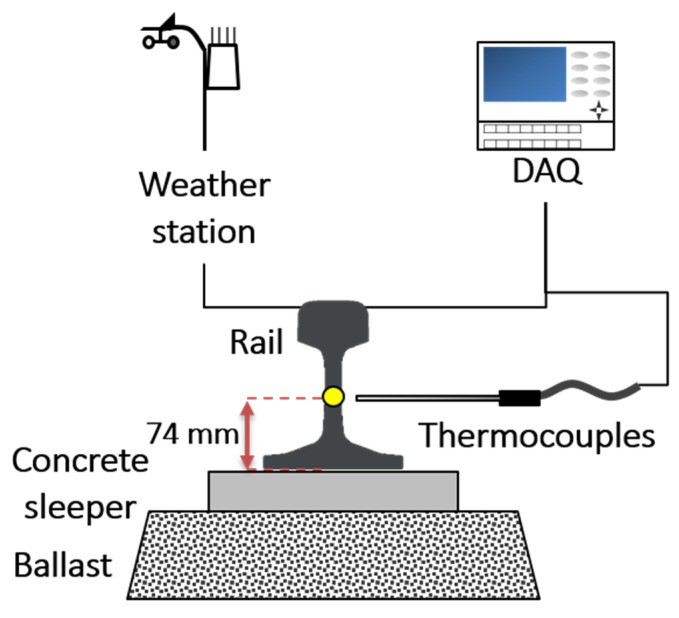
Schematic of the measurement system.

**Figure 2 sensors-21-04606-f002:**
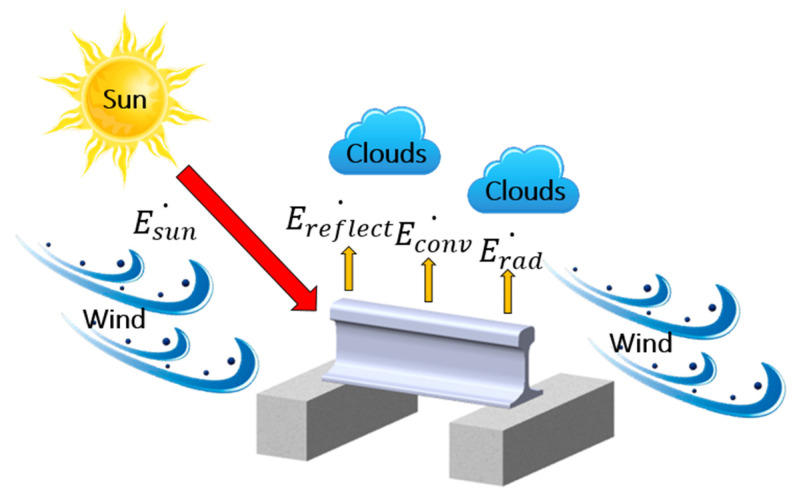
Thermal environment around the rail.

**Figure 3 sensors-21-04606-f003:**
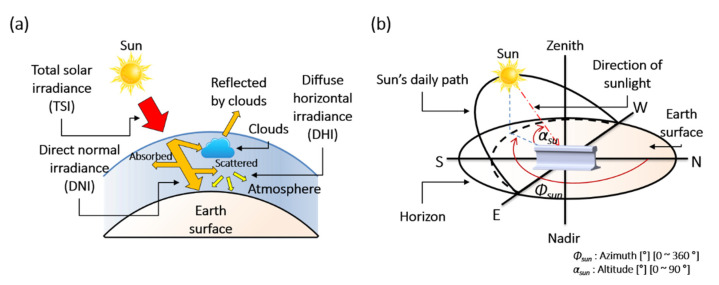
Geometric definitions of (**a**) total solar irradiance (TSI), direct normal irradiance (DNI), and diffuse horizontal irradiance (DHI). (**b**) Azimuth and altitude in the horizontal. coordinate system.

**Figure 4 sensors-21-04606-f004:**
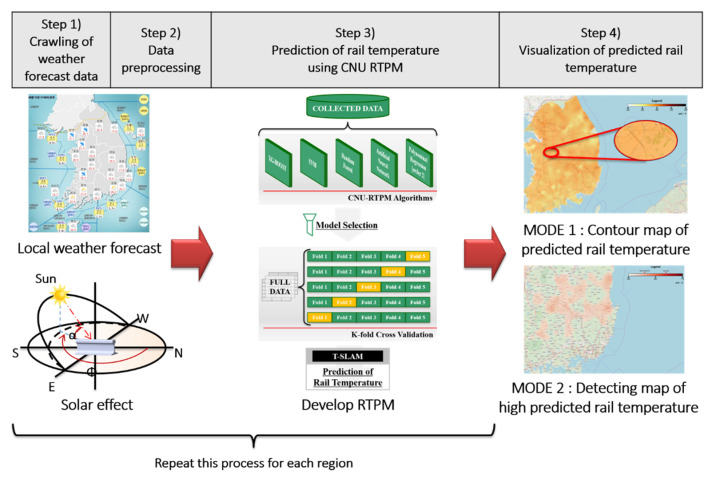
Framework of mapping the local rail-temperature prediction.

**Figure 5 sensors-21-04606-f005:**
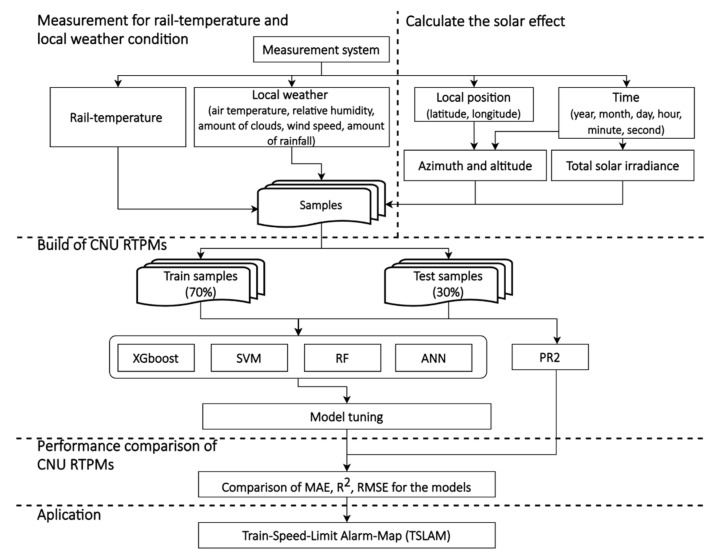
Flowchart of the proposed methodology.

**Figure 6 sensors-21-04606-f006:**
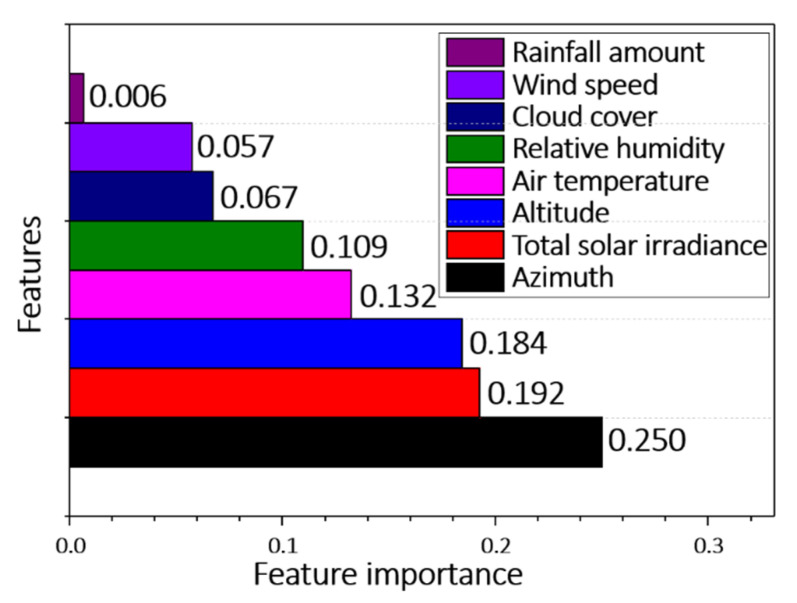
Graded importance of features in the rail-temperature. prediction, obtained by XGBoost.

**Figure 7 sensors-21-04606-f007:**
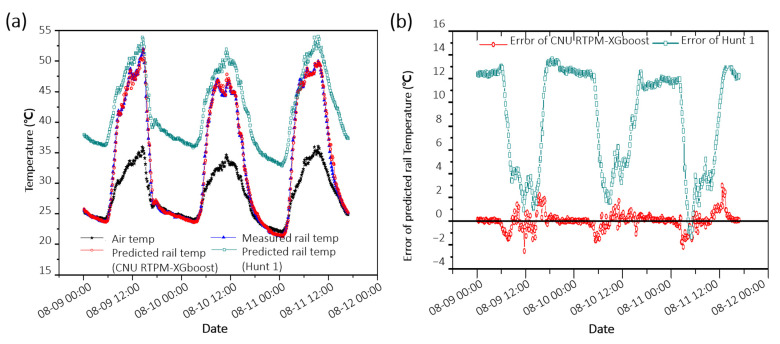
(**a**) Comparison of the predicted rail temperature and the measured rail temperature graphs. (**b**) Error of predicted rail temperature in August 2016.

**Figure 8 sensors-21-04606-f008:**
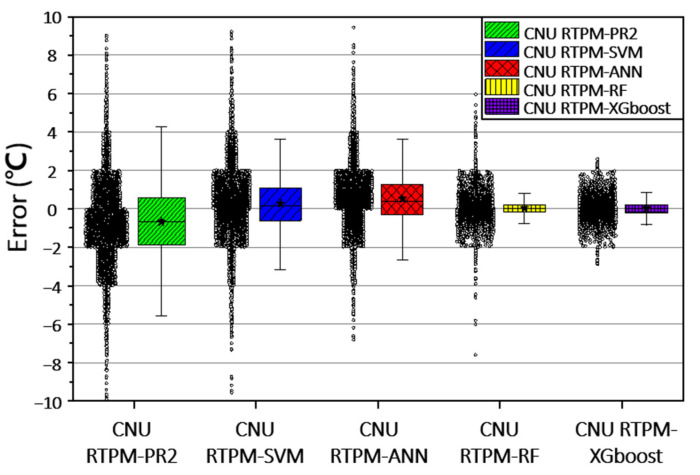
Boxplots showing the error ranges in the CNU RTPMs based on SVM, ANN, PR2, RF, and XGBoost.

**Figure 9 sensors-21-04606-f009:**
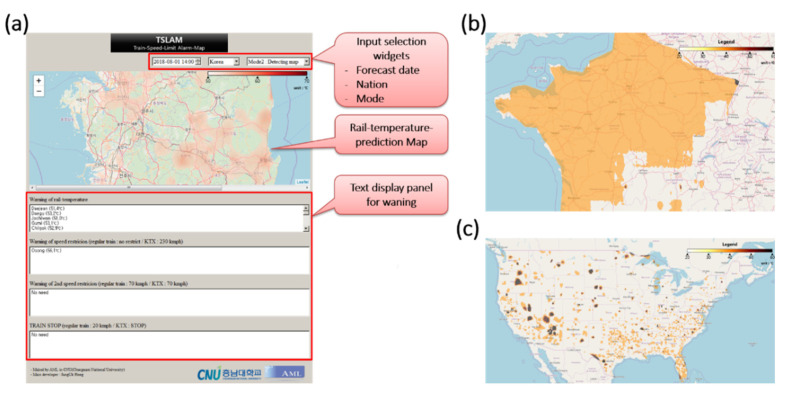
(**a**) GUI of TSLAM and maps of the predicted local rail temperatures in (**b**) France and (**c**) the USA.

**Table 1 sensors-21-04606-t001:** Comparison of prediction errors in previous RTPMs and CNU RTPMs.

Performance Range	Performance in Whole Range of Rail Temperature			Performance in High Rail Temperature Range (over 40 °C)		
Model Name	MAE (°C)	R^2^	RMSE (°C)	MAE (°C)	R^2^	RMSE (°C)
Hunt model 1 [8]	2.913	0.9021	5.866	2.660	0.4471	3.925
Hunt model 2 [8]	14.396	0.9021	15.117	3.661	0.4471	4.448
Munro model [1]	Unknown	0.9180	Unknown	Unknown	Unknown	Unknown
BoM Prediction Equation (1–24 h) [10]	0.136	0.9630	2.560	Unknown	Unknown	Unknown
Weather station Regression model [10]	0.659	0.8960	4.193	Unknown	Unknown	Unknown
CNU RTPM–PR2	0.693	0.9709	2.316	0.103	0.5605	2.7108
Chapman’s model [13]	0.200	Unknown	2.500	Unknown	Unknown	Unknown
CNU Heat Transfer model [7]	1.537	0.9334	3.799	0.618	0.3406	5.935
CNU RTPM–SVM	0.068	0.9720	2.135	1.055	0.5465	2.802
CNU RTPM–ANN	0.519	0.9839	1.732	1.213	0.7115	2.355
CNU RTPM–RF	0.029	0.9972	0.685	0.191	0.9199	0.927
CNU RTPM–XGBoost	0.008	0.9984	0.518	0.119	0.9415	0.771
CNU RTPM–XGBoost Without a solar effect	0.043	0.9816	1.744	0.570	0.7243	1.898

## Data Availability

Not applicable.

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
