# Peer review of "A Rail-Temperature-Prediction Model Based on Machine Learning: Warning of Train-Speed Restrictions Using Weather Forecasting"

_sensors, 2021, doi:10.3390/s21134606_

Round 1
Reviewer 1 Report
The topic of the paper is interesting and suits the Journal . The authors have done a great job on the literature review. However, the introduction needs more attention. Please indicate how many samples for each experiment have been used. Please describe the process of each experiment. Also indicate the model of each tool that is used in the experiment. What is the accuracy of each test? Please explain them accurately.
The discussion part of the paper is too short. This section should be extended and the obtained results should be compared with other approaches and methods. The sensitivity analysis for the applied method should also be added. The conclusions should be re-written and presented in bullets. Besides, the readers expect specific remarks coming from the research. The specific recommendations are also needed.
Author Response
We appreciate your comments. We revised our article and described the details below.
- Please indicate how many samples for each experiment have been used.
- Thanks for your comments on several areas of our manuscript. We were able to indicate the number of samples at each process in the paper. We understand that our paper did not indicate the number of samples used in some processes, which could confound the reader. For example, on page 12, we just showed the percentage of the samples used, not the number. We have revised and added the number of samples used in each process, and we tabulated the details of our revision, as shown below. Also, we highlighted the revised part of the review paper.
* CNU RTPM (Chungnam National University Rail Temperature Prediction Model)
Process |
Page and Line number |
Number of samples |
Measurement, Calculating Solar effect |
Page 12 / line 439 |
35,252 |
Training process at Build of CNU RTPMs* |
Page 12 / line 451 |
70% of the data [24,676] |
Test process at Build of CNU RTPMs* |
Page 12 / line 452 |
30% of the data [10,576] |
Table 1. Details of the indicated the number of samples used at each process
- Please describe the process of each experiment. Also indicate the model of each tools that is used in the experiment
- Proposed methodology on this paper is composed of 4 parts, 1) measurement and calculate the solar feature, 2) build of Chungnam National University Rail Temperature Prediction Model (CNU RTPM), 3) Performance comparison of CNU RTPMs, 4) Application. We described details of the experiment process in figure 5. Also, the details of each process are described from section 2 to section 5 successively.
In addition, we added the library used at building model but didn’t indicate what tools we used. We added the tools we used at building rail temperature prediction model and application at page 7 and page 10. We also highlighted revised part on the review paper.
Page |
Line |
Original |
Revision |
7 |
285-286 |
x |
We built the models using the Python libraries: XGboost, Scikit-learn, Random forest and Tensorflow. |
10 |
387 |
x |
developed by Python |
- What is the accuracy of each test? Please explain them accurately.
- We used three indicators to evaluate the rail temperature prediction models (RTPMs). R-square showed the extent the model explained the dataset, the mean absolute error (MAE) showed the error itself, and the root mean square error (RMSE) showed a more robust error than the MAE. Although MAE is more intuitive in comparing the model performance than RMSE, RMSE is used to evaluate and compare the overall model accuracy because of its better robustness to outliers. Furthermore, RMSE is generally used to evaluate the performance of the rail temperature prediction model. Therefore, RMSE was used mainly as the indicator, whereas MAE and R2 were used as auxiliary indicators for accuracy evaluation. The details of the three indicators are described in Section 4.3 (Comparison methods of model performance (page 9)).
- The discussion part of the paper is too short. This section should be extended and the obtained results should be compared with other approaches and methods.
- We fully agree with your comments and added discussions with various perspectives. First, we discussed the train-speed-limit alarm-map (TSLAM) under the structural health-monitoring perspectives on pages 15 and 16. Also, we highlighted this part in the review paper. TSLAM was developed to support the health condition estimation of rail, referred to as rail temperature prediction, one of the structural health monitoring purposes. This application predicts rail temperature and measures the current rail temperature. Also, the rail temperature could be measured directly using thermocouples. However, thermocouples’ installation over the entire network requires high installation and maintenance costs. However, TSLAM prediction using the real-time forecast can produce nearly the same result as directly measuring the rail temperature. Thus, TSLAM enables the rail temperature indirect measurement.
Second, on page 17, we discussed the factors that affect rail temperature, such as the rail solar absorptivity. The solar absorptivity is important for predicting the rail temperature because solar absorptivity expresses how much solar irradiation affects the temperature change of the rail.
Page |
Line |
Original |
Revision |
15 |
542-546 |
x |
TSLAM is a structural health-monitoring application for analyzing railway safety using CNU RTPM-XGboost, the highest-performing CNU RTPM. Using TSLAM, railway safety managers can know the rail temperature in advance or in real-time. Based on the predicted rail temperature, railway safety managers could decide on safety measures, such as limiting the train speed and spraying water. |
16 |
572-579 |
x |
Additionally, TSLAM can support the indirect measurement of the rail temperature over the entire network. Generally, the rail temperature is directly measured using thermocouples; however, installing thermocouples over the entire network results in high in-stallation and maintenance costs. Also, because of the safety issue, it is difficult to attach the thermocouples in some networks. However, TSLAM prediction showed nearly the same result as directly measuring the rail. Thus, TSLAM is expected to serve as a supporting system that provides the predicted rail temperature using real-time weather measurement data or weather forecast data. |
17 |
603-611 |
x |
Furthermore, the health status of rails can be integrated into an improved RTPM. The health status of the rail is directly related to its thermal properties, such as solar absorptivity, and mechanical properties, which include hardness and density [7,51]. For example, according to a previous study, solar absorptivity is important for predicting rail temperatures [7]. The solar absorptivity expresses how much solar irradiation affects the temperature change of the rail. The rail surface condition affects the solar absorptivity. Usually, the solar absorptivity of unused rail is 98.7%, whereas that of used rail is 81.1%. This difference comes from the rail surface and paint erosion. Thus, the performance of RTPM can be improved in future works by considering the health status of rail. |
- The sensitivity analysis for the applied method should also be added.
- In Section 6.2 (Feature importance), we conducted a feature importance analysis. The feature importance quantitatively evaluates the effect of a particular feature on the model performance. In this section, we analyzed which feature is important or sensitive to the model when predicting the rail temperature. This analysis showed that the solar features (azimuth, total solar irradiance, and altitude) are the most important features in the model. Hence, these features are the most sensitive in predicting the rail temperature.
- The conclusions should be re-written and presented in bullets. Besides, the readers expect specific remarks coming from the research. The specific recommendations are also needed. Can’t understand changing our conclusion part in bullets form.
- Thanks for your insightful comments. We not only revised conclusion section (page 17) in bullet but also added remarks of our paper.
Page |
Line |
Original |
Revision |
17 |
613-634 |
We have developed novel rail-temperature prediction models (CNU RTPMs) and a TSLAM using weather forecast data alone, which predict the rail temperature over the entire network. The CNU RTPMs were built with different machine learning methods using the long-term (over 10 months) measured data from all seasons. Such long-term data col-lection ensures a reliable model. To improve the prediction performance, the CNU RTPMs combine the standard weather features with the solar effect features. These features are easily obtained from global weather forecasts and additional calculations on the weather forecast data at a given time. The newly included solar effect features significantly improved the prediction performance at high rail temperatures. In a performance comparison, CNU RTPM-XGboost emerged as the best predictor of rail temperature among the machine learning methods. Our proposed CNU RTPM-XGboost, which delivered higher performance, reliability, and versatility than previous RTPMs, was suggested as a new model for predicting rail temperature over the entire network. CNU RTPM-XGboost is applicable worldwide, because its features are globally available in weather forecast data. Meanwhile, the visualization application TSLAM maps the predicted rail temperatures, assisting railway safety officers (if necessary) to limit the train speed in advance. We expect that CNU RTPM-XGboost and TSLAM will greatly improve both train safety and train timeliness. |
Herein, we developed novel rail-temperature prediction models (CNU RTPMs) and a TSLAM using weather forecast data alone, which can predict the rail temperature over the entire network. The CNU RTPMs were developed with different machine learning methods using the long-term (over 10 months) measured data from all seasons. Such long-term data collection ensures a reliable model. To improve the prediction performance, the CNU RTPMs combine standard weather features with newly suggested solar effect features. These features originate from the analysis of the thermal environment around the rail. Also, they are easily obtained from global weather forecasts and additional calculations on the weather forecast data. Precisely, the solar effect features significantly improved the prediction performance at the high-rail-temperature range. In a performance comparison, the CNU RTPM-XGboost emerged as the best predictor of the rail temperature among the machine learning methods. The pro-posed CNU RTPM-XGboost, which delivered higher performance, reliability, and versatility than previous RTPMs, was suggested as a new model for predicting rail temperature over the entire network. CNU RTPM-XGboost is applicable worldwide because its features are globally available in weather forecast data. The visualization application, TSLAM, maps the predicted rail temperatures, which assist railway safety officers (if necessary) in planning safety measures. We expect that CNU RTPM-XGboost and TSLAM will significantly improve both train safety and train timeliness. |

Reviewer 2 Report
This study combines hand-made features (or features from empirical physical quantities) with XGBoost to fit the regression formula for temperature prediction. Further improvement is expected as list below.
1. This paper needs to discuss multiple features including material properties and health status of tracks for the models beyond 'sleeper material'. e.g. S Ding, etc., Non-destructive hardness prediction for 18CrNiMo7-6 steel based on feature selection and fusion of Magnetic Barkhausen Noise, NDT & E International 107, 102138, 2019.
2. Could the paper put subsection title e.g. sub-sections 4.1.2, 4.1.3 in full name;
3. More background about structural health monitoring could be discussed in the paper.
Author Response
We appreciate your comments. We are able to check and revise our paper, thanks to your comments. We described the details below.
- This paper needs to discuss multiple features including material properties and health status of tracks for the models beyond 'sleeper material'. e.g. S Ding, etc., Non-destructive hardness prediction for 18CrNiMo7-6 steel based on feature selection and fusion of Magnetic Barkhausen Noise, NDT & E International 107, 102138, 2019.
- We appreciate your insightful comment on our research. We checked the relationship between the health status and the rail temperature. The health status of the rail consists of the thermal property, such as solar absorptivity, and mechanical properties, such as hardness and density. According to a previous study, solar absorptivity is related to the heat transfer in the rail. The rail surface condition affects the solar absorptivity. Usually, the solar absorptivity of an unused rail is 98.7%, whereas that of a used rail is 81.1%. Such a difference can be attributed to the wear on the rail surface and paint erosion. Therefore, we explained the relationship between the health status and the rail temperature in the discussion section (page 17) and reference (reference number is 50).
Page |
Line |
Original |
Revision |
17 |
603-611 |
x |
Furthermore, the health status of rails can be integrated into an improved RTPM. The health status of the rail is directly related to its thermal properties, such as solar absorptivity, and mechanical properties, which include hardness and density [7,51]. For example, according to a previous study, solar absorptivity is important for predicting rail temperatures [7]. The solar absorptivity expresses how much solar irradiation affects the temperature change of the rail. The rail surface condition affects the solar absorptivity. Usually, the solar absorptivity of unused rail is 98.7%, whereas that of used rail is 81.1%. This difference comes from the rail surface and paint erosion. Thus, the performance of RTPM can be improved in future works by considering the health status of rail. |
- Could the paper put subsection title e.g. sub-sections 4.1.2, 4.1.3 in full name.
- Thanks for your comments. We deleted acronyms at title used in paper and write full-name at the title.
- More background about structural health monitoring could be discussed in the paper.
- We agree your comments. According to your comments, we added an additional discussion with structural health monitoring.
- First, we discussed the train-speed-limit alarm-map (TSLAM) under the structural health-monitoring perspectives on pages 15 and 16. Also, we highlighted this part in the review paper. TSLAM was developed to support the health condition estimation of rail, referred to as rail temperature prediction, one of the structural health monitoring purposes. This application predicts rail temperature and measures the current rail temperature. Also, the rail temperature could be measured directly using thermocouples. However, thermocouples’ installation over the entire network requires high installation and maintenance costs. However, TSLAM prediction using the real-time forecast can produce nearly the same result as directly measuring the rail temperature. Thus, TSLAM enables the rail temperature indirect measurement.
Second, on page 17, we discussed the factors that affect rail temperature, such as the rail solar absorptivity. The solar absorptivity is important for predicting the rail temperature because solar absorptivity expresses how much solar irradiation affects the temperature change of the rail.
Page |
Line |
Original |
Revision |
15 |
542-546 |
x |
TSLAM is a structural health-monitoring application for analyzing railway safety using CNU RTPM-XGboost, the highest-performing CNU RTPM. Using TSLAM, railway safety managers can know the rail temperature in advance or in real-time. Based on the predicted rail temperature, railway safety managers could decide on safety measures, such as limiting the train speed and spraying water. |
16 |
572-579 |
x |
Additionally, TSLAM can support the indirect measurement of the rail temperature over the entire network. Generally, the rail temperature is directly measured using thermocouples; however, installing thermocouples over the entire network results in high in-stallation and maintenance costs. Also, because of the safety issue, it is difficult to attach the thermocouples in some networks. However, TSLAM prediction showed nearly the same result as directly measuring the rail. Thus, TSLAM is expected to serve as a supporting system that provides the predicted rail temperature using real-time weather measurement data or weather forecast data. |
17 |
603-611 |
x |
Furthermore, the health status of rails can be integrated into an improved RTPM. The health status of the rail is directly related to its thermal properties, such as solar absorptivity, and mechanical properties, which include hardness and density [7,51]. For example, according to a previous study, solar absorptivity is important for predicting rail temperatures [7]. The solar absorptivity expresses how much solar irradiation affects the temperature change of the rail. The rail surface condition affects the solar absorptivity. Usually, the solar absorptivity of unused rail is 98.7%, whereas that of used rail is 81.1%. This difference comes from the rail surface and paint erosion. Thus, the performance of RTPM can be improved in future works by considering the health status of rail. |

Reviewer 3 Report
Dear Authors
The Reviewer is mostly concerned about the novelty of the work compared to the already published literature. The paper was not written in a good language, the English must be double checked in terms of structure and grammar.
Figures are not good; they are not matched with the merit of the journal. The paper includes too many abbreviations and acronyms without the proper definitions, as a reader, it is hard to track all of them.
The Reviewer is against the publication of the current format of this work due to the following issues:
Serious issues:
- This work is reporting an approach to rail temperature prediction, it is ok, but the Reviewer is not confident about its application for the students and other researchers as a potential reference. It is useful for the industrial applications and partners.
- The authors must be clearer with the methodology definition therefore other researchers could repeat the same procedure.
- The results are simple and there is no complexity to discuss!
General issues:
- Please avoid using the active tense starting with “we” and “our”, it is recommended to change it with the passive tense since it is more suitable for the academic manuscript.
- Figures are not suitable for the academic level.
- In abstract, what do you mean of “we hope that our rail-temperature prediction 19 model will improve track safety and train timeliness.” this is not an academic tense.
- Why the introduction part is so long? The authors must orient the most important aspects of the work by referring some relative references, so introduction part must be rewritten and pointing out the most relevant literature.
- The novelty of the work and its contribution to the state-of-the-art must be better stressed in the abstract or introductory section.
- A table of nomenclature is missing.
Very Best
The Reviewer
Author Response
We appreciate your comments. We revised our article and described the details below.
Serious issue
- This work is reporting an approach to rail temperature prediction, it is ok, but the Reviewer is not confident about its application for the students and other researchers as a potential reference. It is useful for the industrial applications and partners.
- Appreciate your comments. To the best of our knowledge on structural health monitoring, various studies exist that apply machine learning to various academic studies and industrial issues, including application development based on machine learning models. However, most studies only focused on how to build and apply the machine learning model to the issues, whereas others focused on developing applications and their practical uses. We think that both scenarios are essential because engineering aims at practicality. For example, Adam Scianna et al. developed a method to detect global damage in a highway bridge (Ref: Scianna, Adam, et al. “Implementation of a probabilistic structural health monitoring method on a highway bridge.” Advances in Civil Engineering, 2012). He also developed a fully automated application that is combined with a suggested method for practical use. Similarly, our study reports the whole process of how machine learning applies to practical issues, from data acquisition to application development. Such information can help other researchers understand how problems are solved using machine learning and the model’s role in engineering, and consequently, help researchers understand the process and achieve academic progress. Moreover, our study novelly applies machine learning to predict rail temperature using only weather forecasts. Thus, our research can inspire other researchers and serve as an important and potential reference.
- The authors must be clearer with the methodology definition therefore other researchers could repeat the same procedure.
- We totally agree with you that the methodology needs to be revised to enable reproducibility for future related studies. Thus, the methodology was revised more clearly, and specific research process tools were added.
First, the proposed methodology in this study is composed of four parts, which are: 1) measurement and calculation of the solar feature; 2) Development of the Chungnam National University rail temperature prediction model (CNU RTPM); 3) Performance comparison of CNU RTPMs; 4) Application. Figure 5 describes the details of the experimental process. Also, the details of each process are successively described in Sections 2–5.
Second, we indicated the tools used in building the rail temperature prediction model. We stated that the Python library was used in building the model but did not indicate the exact tools. We have added the tools used in building the rail temperature prediction model and its application on pages 8 and 10. We also highlighted the revised part on the review paper.
Finally, we added the appendix section to explain our method. Appendix A describes the summary of the dataset. Appendix B explains the relationship between the global horizontal irradiance and the rail temperature. Appendix C expresses the calculation method of the total solar irradiance. Appendix D dwells on tuning the result of the machine learning models. Appendix E explains the train driving rules according to the rail temperature in Korea.
Page |
Line |
Original |
Revision |
7 |
285-286 |
x |
We built the models using the Python libraries: XGboost, Scikit-learn, Random forest and Tensorflow. |
10 |
387 |
x |
developed by Python |
- The results are simple and there is no complexity to discuss!
- We added discussions with various perspectives. First, we discussed the train-speed-limit alarm-map (TSLAM) under the structural health-monitoring perspectives on pages 15 and 16. Also, we highlighted this part in the review paper. TSLAM was developed to support the health condition estimation of rail, referred to as rail temperature prediction, one of the structural health monitoring purposes. This application predicts rail temperature and measures the current rail temperature. Also, the rail temperature could be measured directly using thermocouples. However, thermocouples’ installation over the entire network requires high installation and maintenance costs. However, TSLAM prediction using the real-time forecast can produce nearly the same result as directly measuring the rail temperature. Thus, TSLAM enables the rail temperature indirect measurement.
Second, on page 17, we discussed the factors that affect rail temperature, such as the rail solar absorptivity. The solar absorptivity is important for predicting the rail temperature because solar absorptivity expresses how much solar irradiation affects the temperature change of the rail.
Page |
Line |
Original |
Revision |
15 |
542-546 |
x |
TSLAM is a structural health-monitoring application for analyzing railway safety using CNU RTPM-XGboost, the highest-performing CNU RTPM. Using TSLAM, railway safety managers can know the rail temperature in advance or in real-time. Based on the predicted rail temperature, railway safety managers could decide on safety measures, such as limiting the train speed and spraying water. |
16 |
572-579 |
x |
Additionally, TSLAM can support the indirect measurement of the rail temperature over the entire network. Generally, the rail temperature is directly measured using thermocouples; however, installing thermocouples over the entire network results in high in-stallation and maintenance costs. Also, because of the safety issue, it is difficult to attach the thermocouples in some networks. However, TSLAM prediction showed nearly the same result as directly measuring the rail. Thus, TSLAM is expected to serve as a supporting system that provides the predicted rail temperature using real-time weather measurement data or weather forecast data. |
17 |
603-611 |
x |
Furthermore, the health status of rails can be integrated into an improved RTPM. The health status of the rail is directly related to its thermal properties, such as solar absorptivity, and mechanical properties, which include hardness and density [7,51]. For example, according to a previous study, solar absorptivity is important for predicting rail temperatures [7]. The solar absorptivity expresses how much solar irradiation affects the temperature change of the rail. The rail surface condition affects the solar absorptivity. Usually, the solar absorptivity of unused rail is 98.7%, whereas that of used rail is 81.1%. This difference comes from the rail surface and paint erosion. Thus, the performance of RTPM can be improved in future works by considering the health status of rail. |
General issue
- Please avoid using the active tense starting with “we” and “our”, it is recommended to change it with the passive tense since it is more suitable for the academic manuscript.
- Thanks for your insightful comments, we fully agree with you that there are too many active tenses and ‘our’ term on our paper. We not only removed and revised our terms but also add some conjunctions. The details are shown below table.
Page |
Line |
Original |
Revision |
2 |
72 |
In our previous RTPM based |
In previous RTPM |
3 |
123-125 |
By minimizing the influence of shadows, we guaranteed a higher temperature in the measurement system than in the shaded area |
By minimizing the influence of shadows, higher temperature is guaranteed in the measurement system. |
4 |
146-147 |
To develop our novel RTPM, we must measure the rail temperature in summer when rail buckling usually occurs. |
To develop our novel RTPM, measuring the rail temperature in summer when rail buckling usually occurs must be needed. |
4 |
155-156 |
To measure the rail temperature at this point, we inserted K-type thermocouple probes by drilling. |
To measure the rail temperature at this point, K-type thermocouple probes were inserted by drilling. |
7 |
281 |
Our developed CNU RTPMs |
CNU RTPMs |
8 |
309-310 |
To minimize the objective function, we add a term to Eq. (3), giving Eq. (5). |
To minimize the objective function, a term is added to Eq. (3), giving Eq. (5). |
9 |
355-356 |
however, we apply 5-fold cross validation to more accurately optimize the hyperparameter combination. |
However, applying 5-fold cross validation is conducted to more accurately optimize the hyperparameter combination |
9 |
364 |
we only present the results of the test data. |
The results of the test data are presented |
9 |
372 |
We computed both measures in our present analysis. |
Both measures are computed in present analysis |
10 |
396 |
Our TSLAM |
TSLAM |
11 |
408-410 |
However, we ultimately crawled the weather forecast data of KMA, which provides de-tailed local weather forecasts over more regions in Korea than the OWM. |
However, Crawling the weather forecast data of KMA, which provides detailed local weather forecast over more regions in Korea the OWM, was selected. |
13 |
466 |
Table 1. Comparison of prediction errors in previous RTPMs and our CNU RTPMs |
Table 1. Comparison of prediction errors in previous RTPMs and CNU RTPMs |
14 |
507-508 |
The ability to predict high rail temperatures was conferred by our selected algorithm and our proposed features |
The ability to predict high rail temperatures was conferred by selected algorithm and proposed features |
14 |
519 |
The practical applicability of our CNU RTPMs |
The practical applicability of CNU RTPMs |
17 |
593 |
We think sleeper material |
Moreover, sleeper material |
- Figures are not suitable for the academic level
- Our purpose of Figure 10 is to help the researchers understand how our rail temperature model and application helps the track health monitoring industries. However, we also understand this figure is not suitable for the academic level and remove the Figure 10.
- In abstract, what do you mean of “we hope that our rail-temperature prediction 19 model will improve track safety and train timeliness.” this is not an academic tense.
- Appreciate your comments, we revised the abstract according to comments.
Page |
Line |
Original |
Revision |
1 |
21-22 |
Combined with TSLAM, we hope that our rail-temperature prediction model will improve track safety and train timeliness. |
Combined with TSLAM, rail-temperature prediction model is expected to improve track safety and train timeliness. |
- Why the introduction part is so long? The authors must orient the most important aspects of the work by referring some relative references, so introduction part must be rewritten and pointing out the most relevant literature.
- We deleted the introduction section. The expression of the specific study method and specific example about the machine learning are omitted from the introduction section.
Page |
Line |
Original |
Revision |
2 |
86-92 |
Recently, machine learning approaches such as artificial neural network (ANN), support vector machine (SVM), random forest (RF), and extreme gradient boosting (XGboost) have attracted great interest by developers of high performance regression models. Machine learning allows a computer to learn the relationship between the data (the input) and the results (the output). For instance, a regression model that predicts the room temperature and daily maximum air temperature by machine learning delivers higher prediction performance than other types of methods. Mateo et al. proposed a machine learning based model that predicts the room temperature from the surrounding climatic conditions and the set room temperature [17]. Abdel-aal et al. proposed a similar model that predicts the maximum daily air temperature from weather data [18]. |
Recently, machine learning approaches such as artificial neural network (ANN), support vector machine (SVM), random forest (RF), and extreme gradient boosting (XGboost) have attracted great interest by developers of high-performance regression models. Machine learning allows a computer to learn the relationship between the data (the input) and the results (the output). For instance, a regression model that predicts the room temperature and daily maximum air temperature by machine learning delivers higher prediction performance than other types of methods [17, 18]. |
3 |
|
The CNU RTPM and its application were built by a sequential process of data acquisition, novel feature design, application of recent machine learning methods, and development of the mapping application. Phase 1: Data measurement. We installed a measurement station that simulates the thermal environment of the rail in the actual field. The rail temperature and weather data were measured every 10 minutes over 10 months. Phase 2: Designing novel features. Using thermal analysis, we analyzed the relationship between the weather factors and the solar effect factors. Both sets of factors were selected as features because they affect the rail temperature. With the selected features, the CNU RTPM can predict the rail temperature over the entire network using weather fore-cast data alone. Especially, the newly introduced solar effect features improve the ability of CNU RTPM to predict high rail temperatures. Phase 3: Applying recent machine learning methods. To build a high performance CNU RTPM, we applied selected features to machine leaning methods (SVM, ANN, RF, and XGboost). As a result, CNU RTPM-XGboost showed the highest performance (R2 = 0.9984, RMSE = 0.518°C), along with higher reliability and broader versatility than previous RTPMs. Phase 4: Developing mapping applications. Our proposed mapping application, called Train-Speed-Limit Alarm-Map (TSLAM), is based on CNU RTPM and visually maps the predicted rail temperature of the entire network for up to 64 hours using weather forecast data alone. Because they require only weather forecast data, our CNU RTPMs are easily combined with various applications. Aided by TSLAM’s visual presentation and provided only with the necessary information, rail safety officers can quickly issue water spraying orders to lower the rail temperature and specify train-speed limits over the entire network. TSLAM is also available worldwide because it uses the global weather forecast data, not merely the data of a particular region. |
x |
- The novelty of the work and its contribution to the state-of-the-art must be better stressed in the abstract or introductory section.
- We modified our research paper according to comments. The key point in this paper, thermal analysis-based feature, is additionally described in abstract and introduction. You can find this page 1 and page 2-3. We also highlight this part on review paper.
Page |
Line |
Original |
Revision |
1 |
9-22 |
Predicting the rail temperature of a railway system is important for establishing a rail management plan against railway derailment caused by orbital buckling. The rail temperature, which is directly responsible for track buckling, is closely related to the air temperature, which is continuously rising under global warming effects. Moreover, to reduce train vibration and noise, railway systems are increasingly installed with continuous welded rails (CWRs). Unfortunately, CWRs are prone to buckling. The present study develops a reliable and highly accurate novel model that predicts the rail temperature by a machine learning method. To predict the rail temperature over the entire network with high prediction performance, we use the weather features and the solar effect features. As a convenient application, we also propose Train-Speed-Limit Alarm-Map (TSLAM), which visually maps the predicted rail-temperature deviations over the entire network for railway safety officers. Combined with TSLAM, we hope that our rail-temperature prediction model will improve track safety and train timeliness. |
Predicting the rail temperature of a railway system is important for establishing a rail management plan against railway derailment caused by orbital buckling. The rail temperature, which is directly responsible for track buckling, is closely related to air temperature, which continuously increases due to global warming effects. Moreover, railway systems are increasingly installed with continuous welded rails (CWRs) to reduce train vibration and noise. Unfortunately, CWRs are prone to buckling. This study develops a reliable and highly accurate novel model that can predict rail temperature using a machine learning method. To predict rail temperature over the entire network with high-prediction performance, the weather effect and solar effect features are used. These features originate from the analysis of the thermal environment around the rail. Precisely, the presented model has a higher performance for predicting high rail temperature than other models. As a convenient structural health-monitoring application, the train-speed-limit alarm-map (TSLAM) was also proposed, which visually maps the predicted rail-temperature deviations over the entire network for railway safety officers. Combined with TSLAM, our rail-temperature prediction model is expected to improve track safety and train timeliness. |
2-3 |
93-105 |
In this study, we propose a machine learning-based RTPM with the highest performance to date (maximum R2 = 0.9984, RMSE = 0.518°C) that predicts the rail temperature over an entire network. The method, called CNU RTPM, outperforms the previous RTPMs in predicting high rail temperatures (over 40°C). Because CNU RTPM uses the weather forecast data alone, it is adaptable to any site that provides a weather forecast. We also develop a mapping application that shows the deviation of the predicted rail temperature over the entire network, enabling quick searching of danger regions. The high performance, versatility, and mapping applications of CNU RTPM are essential for practical use. |
Herein, we propose a machine learning-based rail temperature prediction model (RTPM) with the highest performance to date (maximum R2 = 0.9984, RMSE = 0.518°C) that can predict the rail temperature over an entire network. The method, called Chungnam National University RTPM (CNU RTPM), outperforms the previous RTPMs for predicting high rail temperatures (over 40°C). The CNU RTPM performance is due to the selected features obtained by analyzing the thermal environment around the rail. With these features, the CNU RTPM can predict the rail temperature over the entire network using weather forecast data alone. Also, a structural health-monitoring application, called train-speed-limit alarm-map (TSLAM), was developed, which shows the deviation of the predicted rail temperature over the entire network, enabling quick searching of danger regions. This application is easily combined with the CNU RTPM. TSLAM is also available worldwide because it uses the global weather forecast data, not merely a particular region’s data. |
- A table of nomenclature is missing.
- We apologize that we missed the nomenclature. We add the table of nomenclature at page 18.

Round 2
Reviewer 3 Report
Dear Authors
The revised version is quite fine and at this stage an acceptance is recommended by the Reviewer.
Very Best
The Reviewer